# Association of Affiliate Stigma with Parenting Stress and Its Moderators among Caregivers of Children with Attention-Deficit/Hyperactivity Disorder

**DOI:** 10.3390/ijerph20043192

**Published:** 2023-02-11

**Authors:** Pei-Yun Lin, Wen-Jiun Chou, Ray C. Hsiao, Tai-Ling Liu, Cheng-Fang Yen

**Affiliations:** 1Department of Psychiatry, Kaohsiung Municipal Siaogang Hospital, Kaohsiung 81267, Taiwan; 2Department of Psychiatry, Kaohsiung Medical University Hospital, Kaohsiung Medical University, Kaohsiung 80756, Taiwan; 3Department of Psychiatry, School of Medicine, College of Medicine, Kaohsiung Medical University, Kaohsiung 80708, Taiwan; 4School of Medicine, Chang Gung University, Taoyuan 33302, Taiwan; 5Department of Child and Adolescent Psychiatry, Chang Gung Memorial Hospital, Kaohsiung Medical Center, Kaohsiung 83301, Taiwan; 6Department of Psychiatry and Behavioral Sciences, University of Washington School of Medicine, Seattle, WA 98295, USA; 7Department of Psychiatry, Seattle Children’s, Seattle, WA 98105, USA; 8College of Professional Studies, National Pingtung University of Science and Technology, Pingtung 91201, Taiwan

**Keywords:** affiliate stigma, parenting stress, attention-deficit/hyperactivity disorder, oppositional defiant disorder, psychological well-being

## Abstract

Caring for children with attention-deficit/hyperactivity disorder (CADHD) is stressful for caregivers. Identifying factors related to parenting stress in caregivers of CADHD can facilitate the development of intervention programs. This study aimed to examine the associations between affiliate stigma and various domains of parenting stress among caregivers of CADHD. This study also analyzed the moderating effects of demographic characteristics and the symptoms of childhood ADHD and oppositional defiant disorder (ODD) on the associations between affiliate stigma and parenting stress. In total, 213 caregivers of CADHD participated in this study. Parenting stress was assessed using the Parenting Stress Index, Fourth Edition Short Form (PSI-4-SF). Affiliate stigma was assessed using the Affiliate Stigma Scale. ADHD and ODD symptoms were assessed using the Parent Form of the Swanson, Nolan, and Pelham Scale, Version IV. The results indicated that higher affiliate stigma was significantly associated with greater parenting stress in all three domains of the PSI-4-SF. ODD symptoms increased the magnitude of parenting stress in two domains of parenting stress among caregivers with affiliate stigma. Intervention programs for relieving parenting stress among caregivers of CADHD should take affiliate stigma and child ODD symptoms into consideration.

## 1. Introduction

### 1.1. Parenting Stress in Caregivers of Children with Attention-Deficit/Hyperactivity Disorder

Attention-deficit/hyperactivity disorder (ADHD) is a common childhood neurodevelopmental disorder with a global prevalence between 2% and 7% [1]. The relevant literature demonstrates that caring for children with ADHD (CADHD) is stressful for caregivers and that these caregivers experience more parenting stress than caregivers of children without ADHD. For example, Perez Algorta et al. utilized data from the Multimodal Treatment Study of Children with ADHD (MTA) in the United States (US) to compare parenting stress between 430 mothers of CADHD and 237 mothers of children without ADHD and found that mothers of CADHD had higher parenting stress [2]. Wiener et al. found a higher level of parental stress reported by 84 Canadian parents of adolescents with ADHD compared with 54 parents of non-ADHD adolescents [3]. Cussen et al. compared the level of parenting stress between 30 parents of children that screened positive for ADHD and 156 parents of children that screened negative for ADHD in Australia and found that the former reported a higher level of parenting stress compared with the latter [4]. Theule et al. conducted a meta-analysis to examine the data from 44 studies on a total sample of 4991 families published between 1983 to 2007 and confirmed that parents of CADHD experience more parenting stress than parents of nonclinical controls [5]. A qualitative study also identified parents of CADHD experienced multiple themes of negative impacts of parenting stress on effectively managing CADHD’s behaviors, family relationships, and confidence about one’s own abilities [6]. 

According to Abidin [7], parenting stress can be divided into three domains, namely parental distress (PD), parent–child dysfunctional interaction (PCDI), and a difficult child (DC). The PD domain can be used to assess the level of perceived distress by caregivers due to elements, such as negative emotions, conflicts with a partner or other family members involved in child-rearing, and life interferences, caused by the demands of parenting. The PCDI domain can be used to assess caregivers’ dissatisfaction with the quality of interactions with their children and the degree to which caregivers identify children’s attitudes and behaviors as unacceptable. Finally, the DC domain can be used to assess caregivers’ perceptions of children’s difficulties in self-regulation and the spontaneous fulfillment of their obligations [7]. Studies assessing parenting stress must consider the effects of all three domains.

The factors related to parenting stress are divided into child and parent domains. In the child domain, Wiener et al. found that more severe ADHD and oppositional defiant disorder (ODD) symptoms and other externalizing behaviors were significantly associated with parental stress [3]. The meta-analysis of Theule et al. found that more severe ADHD symptoms and co-occurring conduct problems were significantly associated with parental stress [5]. A study on 80 parents of CADHD in the US found that child aggression, emotional lability, and executive functioning difficulties were significantly associated with parental stress [8]. A study on 13 mothers of girls with ADHD in the US found that girls’ aggression and conduct problems were significantly associated with parental stress [9]. A study on 243 Canadian parents of CADHD aged between 5 and 12 published in 2021 found that CADHD’s executive functional difficulties were significantly associated with parental stress [10]. In the parent domain, the meta-analysis of Theule et al. found that parental depressive symptoms were significantly associated with parental stress [5]. A study on 70 dyads of parents of CADHD in southern Korea demonstrated that maternal anxiety was significantly associated with paternal parenting stress [11]. Wiener et al. found that mothers’ self-reported ADHD symptoms were associated with higher parenting stress, whereas fathers’ self-reported ADHD symptoms were associated with lower parenting stress [3]; the MTA study did not find a significant association between maternal ADHD symptoms and parenting stress [2]. A mediation study on 126 Spanish mothers of CADHD aged 6–17 years old integrated the child and parent factors and found that the association between ADHD and parenting stress was mediated by children’s ADHD and conduct problems and by the negative impact of child ADHD on the family’s social life [12]. Treating the symptoms of children with ADHD, addressing caregivers’ depressive symptoms, and optimizing caregivers’ coping strategies can all reduce parenting stress and improve caregivers’ ability to cope with the difficulties of children with ADHD [13,14,15].

### 1.2. Affiliate Stigma in Caregivers of CADHD

Identifying factors related to parenting stress in caregivers of CADHD can facilitate the development of intervention programs. Affiliate stigma (AS) is a factor related to mental health in caregivers of individuals with psychiatric disorders [16,17]; its role in parenting stress among caregivers of CADHD warrants further examination. AS is internalized self-stigmatization commonly experienced by individuals who care for relatives with psychiatric disorders [16]. Caregivers develop AS through perceived public stigma towards individuals with psychiatric disorders and their caregivers [16]. Caregivers with intense AS may feel shame and embarrassment, develop depression and anxiety, and minimize social interactions to avoid criticism [16,17]. 

In a cross-sectional study on 159 French mothers of boys with ADHD, Charbonnier et al. found that mothers’ distress was positively related with AS [15]. Another cross-sectional study on 400 Taiwanese caregivers of CADHD found that AS is related to caregiver depression [18] and to unfavorable attitudes toward clinical diagnoses, treatment, and etiological explanations for ADHD [19,20,21]. In addition, AS was reported to prospectively predict emotional problems and physical discomfort in Taiwanese CADHD one year later [22]. A study on 63 parents of CADHD in the US also demonstrated that greater AS was associated with more negative parenting [23]. One can reasonably hypothesize that AS is positively associated with parenting stress in caregivers of CADHD; nevertheless, this hypothesis has yet to be formally examined.

Whether ADHD and ODD symptoms moderate the association between AS and parenting stress also warrants investigation. Charbonnier et al. found that boys’ ADHD symptoms were positively associated with maternal AS [15]. A study on Taiwanese caregivers of CADHD found that more severe child inattention symptoms were associated with higher levels of AS [19]. Comorbid ODD is common among CADHD. A review study on six published studies revealed that 1.0% to 13.3% of CADHD have comorbid ODD [24]. CADHD who have ODD symptoms may argue with their parents and refuse to comply with their parents’ requests; ODD symptoms of CADHD may also exacerbate parents’ frustration and low self-confidence in managing CADHD’s behaviors [25]. Therefore, parents of CADHD with co-occurring ODD reported more parenting stress than parents of CADHD without ODD [5]. If ADHD and ODD symptoms moderate the association between AS and parenting stress, intervention programs for improving parenting stress need to consider ADHD and ODD symptoms.

### 1.3. Study Aims

The first aim of this cross-sectional study was to examine the associations between AS and various domains of parenting stress among caregivers of CADHD. The second aim was to analyze the moderating effects of demographic characteristics and the symptoms of childhood ADHD and ODD on the associations between AS and various domains of parenting stress. We proposed the following hypotheses: (1) higher AS is significantly associated with all domains of increased parenting stress in caregivers of CADHD; and (2) demographic characteristics and the symptoms of childhood ADHD and ODD moderate the associations between AS and parenting stress.

## 2. Materials and Methods

### 2.1. Participants and Procedure

We recruited caregivers of CADHD from pediatric mental health outpatient clinics in two general hospitals in Taiwan. The study included caregivers with a child that was aged between 10 and 18 years and was diagnosed as having ADHD in accordance with the DSM-5 criteria [26]. As substance (e.g., methamphetamine [27], opioid [28], and alcohol [29]) use disorders and schizophrenia spectrum and other psychotic disorders [30] increase the risk of cognitive dysfunctions that could potentially impede their understanding of the study aims and procedure, this study excluded caregivers with these psychiatric disorders. Likewise, this study excluded caregivers with intellectual disability. This study also excluded children with other comorbid neurodevelopmental disorders (e.g., intellectual disability, autism spectrum disorder, and learning disorders), mood disorders (e.g., depressive disorders and bipolar spectrum disorder), or schizophrenia spectrum and other psychotic disorders.

This study was conducted between June 2018 and May 2021. Five child psychiatrists from the pediatric mental health outpatient clinics in two general hospitals at Kaohsiung, Taiwan (Kaohsiung Medical University Hospital, with a total of 1720 beds; Kaohsiung Chang Gung Memorial Hospital, with a total of 2680 beds) confirmed the eligibility of 220 caregivers and invited them to participate in the study. In total, 213 caregivers (168 women and 45 men) granted informed consent prior to the assessment and individually completed the self-report questionnaires in research rooms. This study was approved by the institutional review boards of two university hospitals (201800740A3 and KMUHIRB-SV(II)-20170077).

### 2.2. Measures

#### 2.2.1. Parenting Stress Index, Fourth Edition Short Form

We used the Taiwanese version [31] of the Parenting Stress Index, Fourth Edition Short Form (PSI-4-SF) [13] to assess the three domains of caregiver-reported parenting stress (PD, PCDI, and DC). The 36 items on the PSI-4-SF were obtained from the full-length PSI-4. The items on the PD domain assess levels of distress resulting from personal factors of caregivers, such as depression or conflicts with a partner and life restrictions due to the demands of CADHD rearing (e.g., “I often have the feeling that I cannot handle things very well”). The items on the PCDI domain assess levels of caregivers’ dissatisfaction of interactions with CADHD and the degree to which caregivers find CADHD unacceptable (e.g., “My child rarely does things for me that make me feel good”). The items on the DC domain assess CADHD’s self-regulatory abilities perceived by the caregivers (e.g., “My child seems to cry or fuss more often than most children”). The participants rated each item on a 5-point scale with endpoints ranging from 1 (“strongly disagree”) to 5 (“strongly agree”). A higher total subscale score indicates higher parenting stress. The original version of the three domains of the PSI-4-SF had good to satisfactory internal consistency (Cronbach α: 0.88 to 0.90) and acceptable to good test–retest reliability (test–retest coefficient: 0.68 to 0.85) [13]. The Taiwanese version of the PSI-4-SF also had good to satisfactory internal consistency (Cronbach α: 0.86 to 0.91) [31]. The Cronbach α values for the PD, PCDI, and DC domains were 0.91, 0.89, and 0.90, respectively.

#### 2.2.2. Affiliate Stigma Scale

The 22-item Affiliate Stigma Scale (ASS) was used to assess the caregivers’ level of AS toward their child’s ADHD [16]. Each item was rated on a 4-point scale with endpoints ranging from 1 (“strongly disagree”) to 4 (“strongly agree”). A higher total ASS score indicates a higher level of AS. The ASS had satisfactory internal consistency (Cronbach α: 0.94) and predictive validity in a Chinese population in Hong Kong [16] and among caregivers in Taiwan who had relatives with mental illness [32]. The Cronbach α for the ASS in this study was 0.95.

#### 2.2.3. Parent Form of the Swanson, Nolan, and Pelham Scale, Version IV

We invited caregivers to rate the severity of their child’s ADHD and ODD core symptoms in the previous 30 days by using the Chinese version of the Parent Form of the Swanson, Nolan, and Pelham Scale, Version IV (PF-SNAP-IV) [33,34]. The participants rated each item on a 4-point scale with endpoints ranging from 0 (“not at all”) to 3 (“very much”). A higher total subscale score indicates higher levels of inattention, hyperactivity/impulsivity, and ODD core symptoms with the Cronbach α values being 0.89, 0.90, and 0.93.

#### 2.2.4. Demographic Characteristics

We collected data on caregivers’ sex, age, and years of education completed. We also collected data on children’s sex and age.

### 2.3. Statistical Analysis

All statistical analyses were conducted using SPSS 24.0 software (SPSS Inc., Chicago, IL, USA). Categorical variables are presented as percentages, and continuous variables are presented as means and standard deviations (SDs). Multivariate linear regression was used to analyze the associations of AS, demographic characteristics, inattention problems, hyperactive and impulse control problems, and ODD symptoms with the three domains of parenting stress. 

We also used the standard criteria proposed by Baron and Kenny [35] to examine whether the association between AS and parenting stress was different in terms of demographic characteristics, inattention problems, hyperactive and impulse control problems, and ODD symptoms. According to the criteria, moderation occurred when the interaction term for the predictor (AS) and the hypothesized moderator (demographic characteristics, inattention problems, hyperactive and impulse control problems, and ODD symptoms) were significantly associated with the dependent variable (parenting stress) after controlling for the main effects of both the predictors and hypothesized moderator variables. If AS and hypothesized moderators were significantly associated with parenting stress, the interactions (AS × hypothesized moderators) were further incorporated into the regression analysis to examine the moderating effects. Moderation analysis was performed using the PROCESS v4.0.0 macro, which is based on linear regression modeling and developed for Statistical Analysis Software (SAS) 9.4 (SAS Institute Inc., Cary, NC, USA). We considered a *p* value of <0.05 as indicating statistical significance.

## 3. Results

Table 1 presents the demographic characteristics of 213 caregivers (168 women and 45 men; mean age (SD): 44.63 (6.11) years) and their CADHD (32 girls and 181 boys; mean age (SD): 12.88 (2.15) years), in addition to the PF-SNAP-IV, ASS, and PSI-4-SF scores. The mean (SD) values derived for inattention problems, hyperactive and impulse control problems, and ODD symptoms were 13.18 (5.90), 9.01 (5.89), and 9.57 (6.03), respectively. The mean (SD) value derived for AS was 37.84 (10.78). The mean (SD) values derived for parenting stress on the PD, PCDI, and DC domains were 32.54 (9.20), 30.29 (8.52), and 31.90 (8.65), respectively. The absolute skewness and kurtosis values derived for the PSI-4-SF, ASS, and PF-SNAP-IV scores ranged from 0.064 to 0.553 and 0.260 to 0.611, respectively; according to Kim [36], these scores were normally distributed.

Table 2 presents the multivariate linear regression analysis results. The condition index was 29.623, indicating no collinearity as determined on the basis of the suggestions of Senaviratna and Cooray [37]. The results revealed that AS was significantly associated with increased parenting stress in all three domains of the PSI-4-SF. Inattention problems were also significantly associated with increased parenting stress in the PD and DC domains. Significantly positive associations were detected between child ODD symptoms and increased parenting stress in the PCDI and DC domains.

Since inattention problems and ODD symptoms had significant associations with parenting stress, we used their interactions with AS as inputs in our multivariate linear regression analysis model to examine whether inattention problems and ODD symptoms moderated the association between AS and parenting stress (Table 3). The results revealed that the interaction between ODD symptoms and AS increased the magnitude of parenting stress in the PCDI (coefficient = 0.018, *p* < 0.05) and DC domains (coefficient = 0.020, *p* < 0.01), indicating that ODD symptoms moderated the associations between AS and parenting stress in the PCDI and DC domains. The moderating effects are presented in Figure 1. Higher levels of ODD symptoms corresponded to higher parenting stress in the PCDI and DC domains among participants with AS.

## 4. Discussion

### 4.1. Parenting Stress and AS

This study demonstrated that after demographic characteristics and childhood ADHD and ODD symptoms were controlled for, AS was positively associated with all three domains of parenting stress in caregivers of CADHD. This study is the first one to examine the associations between AS and various domains of parenting stress in caregivers of CADHD. Although the cross-sectional design limited the possibility to determine the casual relationship between AS and parenting stress, there can be reciprocal relationship between them. Parenting stress arises when parenting demands exceed the expected and actual resources available to the parents that permit them to succeed in the parent role [13]. Stressful feelings and experiences of failure in parenting will discourage caregivers of CADHD and increase the risk of internalizing public stigma toward ADHD as their attitudes toward their CADHD. Alternatively, AS may increase multiple domains of parenting stress. Caregivers who have higher parenting stress in the PD domain perceived greater distress due to negative emotions, conflicts with family members, and life interferences caused by the demands of parenting [13]. Charbonnier et al. observed that AS was positively correlated with depression and anxiety in mothers of CADHD and was negatively correlated with mothers’ life and self-satisfaction levels [15]. AS may compromise caregivers’ mental health and increase caregivers’ difficulties in cooperating with family members involved in child-rearing; then, the parenting stress in the PD domain increases. Caregivers with higher parenting stress in the PCDI domain had greater dissatisfaction with the quality of interactions with their CADHD and greater disapproval towards CADHD’s behaviors [13]. Research found that greater AS was associated with more negative parenting among caregivers of CADHD [23]. AS may reduce caregivers’ patience and ability to learn how to well interact with their CADHD and further exacerbate parent–child dysfunctional interactions. Caregivers with higher parenting stress in the DC domain perceived greater CADHD’s difficulties in self-regulation and fulfillment of their obligations [13]. Studies have found that AS is related to caregivers’ unfavorable attitudes toward clinical diagnoses and treatment for ADHD [19,20,21]. The timing of proper treatment for ADHD may be delayed, and children’s ADHD symptoms and difficulties in daily lives could be aggravated. Research found that caregivers’ AS predicted CADHD’s emotional problems and physical discomfort [22]; therefore, caregivers’ parenting stress in the DC domain may increase. Caregivers’ stress-coping and child-rearing styles affect the level of affiliate stigma, for example, caregivers with higher care and affection parenting had lower affiliate stigma [38]. Intervention programs are necessary to diminish AS and parenting stress in caregivers of CADHD.

### 4.2. Moderating Effect of ODD Symptoms

Previous studies have revealed that more severe ODD symptoms and disturbed behaviors in children were positively associated with higher parenting distress in caregivers of CADHD [3,39]. The parents of children diagnosed as having ODD were more psychologically vulnerable compared with the parents of children diagnosed as having generalized anxiety disorder [40]. Ross et al. also noted that mothers of children who received dual diagnoses of ADHD and ODD reported a higher level of stress rooted in the interactions with their children than did mothers of children with ADHD or ODD alone [41]. Furthermore, children with ODD often report stigmatizing experiences that reduce their willingness to attend school and increase their risk of placement in the juvenile justice system and risk of receiving unfair sentences [42,43,44]. However, research found that ODD symptoms of CADHD had various associations with the three domains of parenting stress. Wiener et al. found that mothers of CADHD with ODD reported higher levels of parenting stress than mothers of CADHD without ODD in the DC and PCDI domains and that fathers of CADHD with ODD reported more PCDI domain stress than fathers of CADHD without ODD [3]. The present study also found that higher ODD symptoms of CADHD were significantly associated with parenting stress in the DC and PCDI but not PD domains and further moderated the association between AS and parenting stress in the PCDI and DC domains. Children with ODD symptoms are uncooperative, defiant, and hostile toward caregivers and have low intentions to comply with caregivers’ requests. Therefore, parenting stress in the DC and PCDI domains involving the management of CADHD’s behaviors can increase [25]. Since ODD is commonly comorbid with ADHD [45,46,47] and is an indicator of early-onset conduct disorder [48], symptoms of ODD in children can strengthen the association between AS and parenting stress in caregivers of CADHD.

### 4.3. Implications for Practice

We recommend the implementation of intervention programs targeting AS and parenting stress. Inadequate public knowledge of ADHD may aggravate social stigmatization and AS. The promotion of psychoeducation to increase the public’s knowledge of ADHD may mitigate social stigmatization and AS [49,50]. Equipping parents with coping strategies, such as positive reframing, enhancing the effectiveness of parent management training, providing adequate social support, and reducing the symptoms of ADHD, may help alleviate parenting stress [8,13,51,52]. Moreover, considering the moderating effect of ODD symptoms on the association between AS and parenting stress, such symptoms warrant early diagnosis and intervention [45]. Customizing treatment strategies for ODD according to various biological and social factors, such as the sex of the child and the mothers’ educational level [53], may help reduce the severity of ODD symptoms and thus modify the relationship between AS and parenting stress.

### 4.4. Limitations

This study has several limitations. First, because we collected all information from a single source (caregivers), common-method variance may be present. Second, the causality between AS and parenting stress could not be determined. Third, the study did not analyze psychosocial factors, such as support systems and parents’ coping skills. These factors may have influenced the association between AS and parenting stress. Strategies that caregivers used to cope with parenting stress were not surveyed. Fourth, this study recruited caregivers of CADHD from the outpatient clinics. Whether the results of this study can be generalized to caregivers who did not take their CADHD to child psychiatrists warrants further study.

## 5. Conclusions

AS was positively associated with parenting stress in caregivers of CADHD. Children with ODD symptoms increased the magnitude of parenting stress in the PCDI and DC domains in participants with AS. Intervention programs targeting AS and parenting stress are warranted to enhance the mental health of caregivers of CADHD. As a moderator of the association between AS and parenting stress, ODD symptoms warrant early identification and intervention.

## Figures and Tables

**Figure 1 ijerph-20-03192-f001:**
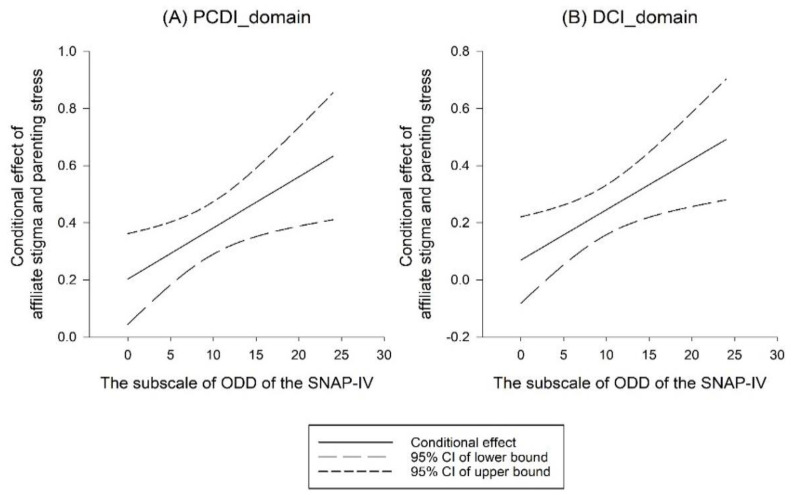
Moderation effects of oppositional defiant disorder (ODD) symptoms on the association between affiliate stigma and parenting stress in caregivers. Mean centering was performed on the predictor (affiliate stigma) and moderator (ODD) symptoms) prior to moderation analysis to enhance comparability.

**Table 1 ijerph-20-03192-t001:** Caregiver and Child Factors and Parenting Stress (N = 213).

	*n* (%)	Mean (SD)	Range
*Caregiver factors*			
Sex			
Female	168 (78.9)		
Male	45 (21.1)		
Age (years)		44.63 (6.11)	30–69
Years of education (years)		14.15 (3.02)	6–24
Affiliate stigma		37.84 (10.78)	22–72
*Children factors*			
Sex			
Boy	181 (85.0)		
Girl	32 (15.0)		
Age (years)		12.88 (2.15)	10–18
Inattention		13.18 (5.90)	1–27
Hyperactivity/impulsivity		9.01 (5.89)	0–26
Oppositional defiance		9.57 (6.03)	0–24
*Parenting stress*			
Parental distress		32.54 (9.20)	12–57
Parent–child dysfunctional interaction		30.29 (8.52)	12–52
Difficult child		31.90 (8.65)	11–54

**Table 2 ijerph-20-03192-t002:** Associations of Child and Caregiver Factors and Affiliate Stigma with Parenting Stress: Multivariate Linear Regression.

	Parental Distress	Parent–Child Dysfunctional Interaction	Difficult Child
	*B (se)*	*B (se)*	*B (se)*
Caregivers’ sex ^a^	−1.297 (1.386)	2.131 (1.208)	1.348 (1.150)
Caregivers’ age	−0.027 (0.098)	0.119 (0.085)	0.052 (0.081)
Caregivers’ education	0.068 (0.190)	−0.322 (0.165)	−0.068 (0.157)
Children’s sex ^b^	1.104 (1.592)	0.478 (1.387)	−0.666 (1.321)
Children’s age	−0.231 (0.284)	0.172 (0.248)	−0.035 (0.236)
Children’s inattention	0.247 (0.118) *	0.147 (0.103)	0.201 (0.098) *
Children’s hyperactivity/impulsivity	−0.033 (0.136)	−0.105 (0.118)	−0.028 (0.113)
Children’s oppositional defiance	0.165 (0.122)	0.286 (0.106) **	0.669 (0.101) ***
Affiliate stigma	0.340 (0.054) ***	0.371 (0.047) ***	0.235 (0.045) ***
R^2^	0.256	0.342	0.421
Adjusted R^2^	0.223	0.312	0.395
*F*	7.780 ***	11.698 ***	16.400 ***

^a^: females as the reference; ^b^: girls as the reference. * *p* < 0.05, ** *p* < 0.01, *** *p* < 0.001.

**Table 3 ijerph-20-03192-t003:** Moderating Effects of Children’s Inattention and Oppositional Defiance on the Association between Affiliate Stigma and Parenting Stress: Multivariate Linear Regression.

	Parental Distress	Parent–Child Dysfunctional Interaction	Difficult Child
	*B (se)*	*B (se)*	*B (se)*
Caregivers’ age	−0.029 (0.098)	0.119 (0.084)	0.057 (0.080)
Caregivers’ education	0.065 (0.190)	−0.354 (0.163) *	−0.094 (0.156)
Children’s sex	1.099 (1.596)	0.574 (1.369)	−0.548 (1.303)
Children’s age	−0.242 (0.286)	0.119 (0.245)	−0.066 (0.234)
Caregivers’ sex	−1.299 (1.389)	2.566 (1.204) *	1.839 (1.147)
Children’s inattention	0.128 (0.346)	0.196 (0.103)	0.556 (0.302)
Children’s hyperactivity/impulsivity	−0.035 (0.136)	−0.155 (0.118)	−0.079 (0.112)
Children’s oppositional defiance	0.170 (0.123)	−0.398 (0.289)	−0.106 (0.290)
Affiliate stigma	0.298 (0.128) *	0.203 (0.081) *	0.155 (0.111)
Inattention X Affiliate stigma	0.003 (0.009)		−0.008 (0.007)
Oppositional defiance X Affiliate stigma		0.018 (0.007) *	0.020 (0.007) **

* *p* < 0.05, ** *p* < 0.01.

## Data Availability

The data used in this study are available upon reasonable request to the corresponding authors.

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
