# Peer review of "Association of Affiliate Stigma with Parenting Stress and Its Moderators among Caregivers of Children with Attention-Deficit/Hyperactivity Disorder"

_ijerph, 2023, doi:10.3390/ijerph20043192_

Round 1
Reviewer 1 Report
Thank you for the opportunity to review this article.
The study examines the Association of Affiliate Stigma with Parenting Stress and Its Moderators Among Caregivers of Children with Attention-Deficit/Hyperactivity Disorder with a look on coexistent ODD symptoms.
The article is clearly written, the introduction is a sufficient explanation of the current data.
Methodology is accurate, inclusion and exclusion criteria are well chosen. Diagnosis of ADHD is confirmed by clinicians using established criteria.
Results are clearly presented, and the discussion, pointing to the scale of CADHD burden and implications for practice is sufficient.
Few limitations are mentioned.
The article is acceptable for publishing as is. My only reservation is the little originality of the paper and its conclusions.
Author Response
We appreciated your valuable comments. As discussed below, we have revised our manuscript with underlines based on your suggestions. Please let us know if we need to provide anything else regarding this revision.
Comment 1
Few limitations are mentioned.
Response
Thank you for your comment. We added more limitations of this study as below. Please refer to line 345-349.
“Third, ...Strategies that caregivers used to cope with parenting stress were not surveyed.”
“Fourth, this study recruited caregivers of CADHD from the outpatient clinics. Whether the results of this study can be generalized to caregivers who did not take their CADHD to child psychiatrists warrants further study.”
Comment 2
The article is acceptable for publishing as is. My only reservation is the little originality of the paper and its conclusions.
Response
Thank you for your comment. We searched for previous studies on the same topic in PubMed by using “ADHD,” “parenting stress,” and “affiliate stigma” and found only one study has examined this issue (Charbonnier E, Caparos S, Trémolière B. The role of mothers' affiliate stigma and child's symptoms on the distress of mothers with ADHD children. J Ment Health. 2019 Jun;28(3):282-288. doi: 10.1080/09638237.2018.1521944.). We believe that this study will add knowledge to the field of ADHD studies.
Reviewer 2 Report
Dear Authors
After the review of this document, I noticed that this draft was carefully written, the introduction appropriately problematizes and justify the present study. In the same way the methods and instruments are suitable to test the hypothesis, and the discussion clearly presents the balance between theory and practice.
It would have been interesting if from the design of the study some of these validity threats had been considered in order to have greater certainty of the results, for example, to have considered some assessment for the coping mechanisms of the parents with respect to distress.
Author Response
We appreciated your valuable comments. As discussed below, we have revised our manuscript with underlines based on your suggestions. Please let us know if we need to provide anything else regarding this revision.
Comment 1
It would have been interesting if from the design of the study some of these validity threats had been considered in order to have greater certainty of the results, for example, to have considered some assessment for the coping mechanisms of the parents with respect to distress.
Response
Thank you for your comment. We added it as one of the limitations of this study as below. Please refer to line 345.
“Third, ...Strategies that caregivers used to cope with parenting stress were not surveyed.”
Reviewer 3 Report
Title: Is there a hyphen in "Attention Deficit/Hyperactivity Disorder"? Because in the text there isn't, but in the title there is.
Abstract: This sentence is rather long (ll.23-27). May I suggest that the aims are separated into two sentences?
l. 27: "participated into this study" should be "participated in this study"
ll. 30-31: "The results 30 indicated that affiliate stigma..." - may I know if it is higher or lower affiliate stigma?
Introduction. ll.42-45: "The relevant literature" - it is not mentioned where the studies were held or in what context these were done. If a systematic review and meta-analysis, kindly state how many studies, were they global studies, and the date range of the articles were published.
Could the authors add something as to why childhood ODD symptoms are included? Yes, a few studies were cited, but it is very thin justification. Perhaps one paragraph elucidating ODD and ADHD comorbidity, how ODD is associated with affiliate stigma, and how they have been related in past studies on parental stress would be interesting. Again, I believe the context of the studies should be mentioned in the text.
Some of the studies cited are really dated. May I know why that is the case? Is it possible to cite more recent studies?
For hypothesis 1 (ll.89-90), "AS is significantly associated..." Did the authors mean "Higher AS is significantly associated..."
Methods. ll. 98-99, the authors mentioned "The study excluded caregivers with any cognitive problems (e.g., those with substance-use disorders...". I don't think substance-use disorders is a cognitive problem, and neither is schizophrenia.
More information is needed on where the five child psychiatrists practice. Are they from different sites or from the same hospital, and what locality is/are the hospital (s) or clinics (s)?
Under measures, could you explain a bit more about the three domains of parenting stress? It is not clear in the introduction nor in the methods section what these domains are about.
Under statistical analysis, may I know how you determined whether there was a moderating effect of demographic characteristics, inattention problems, hyperactive and impulse control problems, and ODD symptoms? Please clarify clearly how the moderation analysis was done, and how the effect of moderation was determined.
Results. In Table 2, please indicate which is the reference group, male or female, yes or no, for each of the categorical variables used.
Please report the R-squared, adjusted R squared values, and the F statistics of each multiple regression model.
l. 167 - "AS was significantly associated" did you mean "higher AS was significantly associated..."
l.180, rather than "increased parenting stress", kindly consider changing it to "increased the magnitude of parenting stress". It would be good to indicate how much increase there was.
More elaboration on the results for table 3, especially to provide results on by parenting stress domains (the three domains of parental distress, parent-child dysfunctional interaction, and difficult child.
Discussion. The discussion section is too short for the interpretation to be meaningful. For example, I see that the analysis involved three separate domains of parenting stress; however, this was not discussed in the discussion section. Rather parenting stress was treated as a unidimensional variable. Kindly involve a discussion on how different domains of parenting stress was affected by AS. It is interesting that ODD was not associated with parental distress, but yes for the other two domains of parenting stress.. Why do you think that is the case, and how is that different from past literature? Similarly, I would like to suggest discussing the moderating effect of the various factors tested on the three different domains of parenting stress, since different results were obtained for different parenting stress domains.
Some of the studies cited are really dated. May I know why that is the case?
Under limitations, please specify how the generalizability of the study is affected by the recruitment method.
Author Response
We appreciated your valuable comments. As discussed below, we have revised our manuscript with underlines based on your suggestions. Please let us know if we need to provide anything else regarding this revision.
Comment 1
Title: Is there a hyphen in "Attention Deficit/Hyperactivity Disorder"? Because in the text there isn't, but in the title there is.
Response
Thank you for your reminding. We changed it into "Attention-Deficit/Hyperactivity Disorder" in the text. Please refer to lines 21, 40 and 41.
Comment 2
Abstract: This sentence is rather long (ll.23-27). May I suggest that the aims are separated into two sentences?
Response
We changed the aims into two sentences as below. Please refer to line 23-27.
“This study aimed to examine the associations between affiliate stigma and various domains of parenting stress among caregivers of CADHD. This study also analyzed the moderating effects of demographic characteristics and the symptoms of childhood ADHD and oppositional defiant disorder (ODD) on the associations between affiliate stigma and parenting stress.”
Comment 3
- 27: "participated into this study" should be "participated in this study"
Response
We changed it into “participated in this study.” Please refer to line 27.
Comment 4
- 30-31: "The results 30 indicated that affiliate stigma..." - may I know if it is higher or lower affiliate stigma?
Response
We added “higher” into this sentence. Please refer to line 31.
Comment 5
Introduction. ll.42-45: "The relevant literature" - it is not mentioned where the studies were held or in what context these were done. If a systematic review and meta-analysis, kindly state how many studies, were they global studies, and the date range of the articles were published.
Response
We added explanations for the studies cited.
“For example, Perez Algorta et al. utilized data from the Multimodal Treatment Study of Children with ADHD (MTA) in the United States (US) to compare parenting stress between 430 mothers of CADHD and 237 mothers of children without ADHD and found that mothers of CADHD had higher parenting stress [2]. Wiener et al. found a higher level of parental stress reported by 84 Canadian parents of adolescents with ADHD compared with 54 parents of non-ADHD adolescents [3]. Cussen et al. compared the level of parenting stress between 30 parents of children screened positive for ADHD and 156 parents of children screened negative for ADHD in Australia found that the former reported a higher level of parenting stress compared with the latter [4]. Theule et al. conducted a meta-analysis to examine the data from 44 studies on a total sample of 4,991 families published between 1983 to 2007 and confirmed that parents of CADHD experience more parenting stress than parents of nonclinical controls [5]. A qualitative study also identified parents of CADHD experienced multiple themes of negative impacts of parenting stress on effectively managing CADHD’s behaviors, family relationship, and confidence about one's own abilities [6].” Please refer to line 45-59.
“Wiener et al. found that more severe ADHD and oppositional defiant disorder (ODD) symptoms and other externalizing behaviors were significantly associated with parental stress [3]. The meta-analysis of Theule et al. found that more severe ADHD symptoms and co-occurring conduct problems were significantly associated with parental stress [5]. A study on 80 parents of CADHD in the US found that child aggression, emotional lability, and executive functioning difficulties were significantly associated with parental stress [8]. A study on 13 mothers of girls with ADHD in the US found that girls’ aggression and conduct problems was were significantly associated with parental stress [9]. A study on 243 Canadian parents of CADHD aged between 5 to 12 published in 2021 found that CADHD’s executive functional difficulties were significantly associated with parental stress [10]. In the parent domain, the meta-analysis of Theule et al. found that parental depressive symptoms were significantly associated with parental stress [5]. A study on 70 dyads of parents of CADHD in southern Korea demonstrated that maternal anxiety was significantly associated with paternal parenting stress [11]. Wiener et al. found that mothers’ self-reported ADHD symptoms were associated with higher parenting stress, whereas fathers’ self-reported ADHD symptoms were associated with lower parenting stress [3]; the MTA study did not find a significant association between maternal ADHD symptoms and parenting stress [2]. A mediation study on 126 Spanish mothers of CADHD aged 6-17 years old integrated the child and parent factors and found that the association between ADHD and parenting stress was mediated by children's ADHD and conduct problems and by negative impact of child ADHD on family's social life [12].” Please refer to line 72-93.
“In a cross-sectional study on 159 French mothers of boys with ADHD, Charbonnier et al. found that mothers’ distress was positively related with AS [15]. Another cross-sectional study on 400 Taiwanese caregivers of CADHD found that AS is related to caregiver depression [18] and to unfavorable attitudes toward clinical diagnoses, treatment, and etiological explanations for ADHD [19–21]. In addition, AS was reported to prospectively predict emotional problems and physical discomfort in Taiwanese CADHD one year later [22]. A study on 63 parents of CADHD in the US also demonstrated that greater AS was associated with more negative parenting [23].” Please refer to line 107-114.
Comment 6
Could the authors add something as to why childhood ODD symptoms are included? Yes, a few studies were cited, but it is very thin justification. Perhaps one paragraph elucidating ODD and ADHD comorbidity, how ODD is associated with affiliate stigma, and how they have been related in past studies on parental stress would be interesting. Again, I believe the context of the studies should be mentioned in the text.
Response
Thank you for your suggestions. We added a new paragraph to introduce the roles of ADHD and ODD symptoms on parenting stress and affiliate stigma in caregivers of children with ADHD as below. Please refer to line 117-129.
“Whether ADHD and ODD symptoms moderate the association between AS and parenting stress also warrants investigation. Charbonnier et al. found that boys’ ADHD symptoms was positively associated with maternal AS [15]. A study on Taiwanese caregivers of CADHD found that more severe child inattention symptoms were associated with higher levels of AS [19]. Comorbid ODD is common among CADHD. A review study on six published studies revealed that 1.0% to 13.3% of CADHD have comorbid ODD [24]. CADHD who have ODD symptoms may argue with their parents and refuse to comply with their parents’ requests; ODD symptoms of CADHD may also exacerbate parents’ frustration and low self-confidence in managing CADHD’s behaviors [25]. Therefore, parents of CADHD with co-occurring ODD reported more parenting stress than parents of CADHD without ODD [5]. If ADHD and ODD symptoms moderate the association between AS and parenting stress, intervention programs for improving parenting stress need to consider ADHD and ODD symptoms.”
Comment 7
Some of the studies cited are really dated. May I know why that is the case? Is it possible to cite more recent studies?
Response
Indeed, some studies cited in this study were published a decade ago; however, they were classic works on parenting stress among parents of children with ADHD (e.g., Breen and Barkley, 1988). Therefore, we kept them in the revised manuscript. We also cited more studies published in recent years in the revised manuscript.
- McLuckie, A.; Landers, A.L.; Rowbotham, M.; Landine, J.; Schwartz, M.; Ng, D. Are parent- and teacher-reported executive function difficulties associated with parenting stress for Children diagnosed with ADHD? J Atten Disord. 2021, 25, 22-32. doi: 10.1177/1087054718756196.
- Lee, Y.J.; Kim, J. Effect of maternal anxiety on parenting stress of fathers of children with ADHD. J Korean Med Sci. 2022, 37, e89. doi: 10.3346/jkms.2022.37.e89.
- Muñoz-Silva, A.; Lago-Urbano, R.; Sanchez-Garcia, M.; Carmona-Márquez, J. Child/adolescent's ADHD and parenting stress: The mediating role of family impact and conduct problems. Front Psychol. 2017, 8, 2252. doi: 10.3389/fpsyg.2017.02252.
- Mikami, A.Y.; Chong, G.K.; Saporito, J.M.; Na, J.J. Implications of parental affiliate stigma in families of children with ADHD. J. Clin. Child Adolesc. Psychol. 2015, 44, 595–603. doi:10.1080/15374416.2014.888665.
- Hsieh, Y.P.; Wu, C.F.; Chou, W.J.; Yen, C.F. Multidimensional correlates of parental self-efficacy in managing adolescent internet use among parents of adolescents with attention-deficit/hyperactivity disorder. Int J Environ Res Public Health. 2020, 17, 5768. doi:10.3390/ijerph17165768.
- Maughan, B.; Rowe, R.; Messer, J.; Goodman, R.; Meltzer, H. Conduct disorder and oppositional defiant disorder in a national sample: Developmental epidemiology. J Child Psychol Psychiatry. 2004, 45, 609–621. doi: 10.1111/j.1469-7610.2004.00250.x.
- Potvin S, Pelletier J, Grot S, Hébert C, Barr AM, Lecomte T. Cognitive deficits in individuals with methamphetamine use disorder: A meta-analysis. Addict Behav. 2018 May;80:154-160. doi: 10.1016/j.addbeh.2018.01.021.
- Wollman SC, Hauson AO, Hall MG, Connors EJ, Allen KE, Stern MJ, Stephan RA, Kimmel CL, Sarkissians S, Barlet BD, Flora-Tostado C. Neuropsychological functioning in opioid use disorder: A research synthesis and meta-analysis. Am J Drug Alcohol Abuse. 2019;45(1):11-25. doi: 10.1080/00952990.2018.1517262.
- Rehm J, Hasan OSM, Black SE, Shield KD, Schwarzinger M. Alcohol use and dementia: a systematic scoping review. Alzheimers Res Ther. 2019 Jan 5;11(1):1. doi: 10.1186/s13195-018-0453-0.
- Khalil M, Hollander P, Raucher-Chéné D, Lepage M, Lavigne KM. Structural brain correlates of cognitive function in schizophrenia: A meta-analysis. Neurosci Biobehav Rev. 2022 Jan;132:37-49. doi: 10.1016/j.neubiorev.2021.11.034.
Comment 8
For hypothesis 1 (ll.89-90), "AS is significantly associated..." Did the authors mean "Higher AS is significantly associated..."
Response
We added “Higher” into this sentence. Please refer to line 135.
Comment 9
Methods. ll. 98-99, the authors mentioned "The study excluded caregivers with any cognitive problems (e.g., those with substance-use disorders...". I don't think substance-use disorders is a cognitive problem, and neither is schizophrenia.
Response
Thank you for your comment. We added the explanations based on the results of previous meta-analyses for why this study excluded caregivers who had substance use disorder and schizophrenia spectrum and other psychotic disorders as below. Please refer to line 144-149.
“Because that substance (e.g., methamphetamine [27], opioid [28], and alcohol [29]) use disorders and schizophrenia spectrum and other psychotic disorders [30] increase the risk of cognitive dysfunctions that could potentially impede their understanding of the study aims and procedure, this study excluded caregivers with these psychiatric disorders. Likewise, this study excluded caregivers with intellectual disability.”
Comment 10
More information is needed on where the five child psychiatrists practice. Are they from different sites or from the same hospital, and what locality is/are the hospital (s) or clinics (s)?
Response
We added information regarding where the five child psychiatrists practiced as below. Please refer to line 153-156.
“Five child psychiatrists from the pediatric mental health outpatient clinics in two general hospitals at Kaohsiung, Taiwan (Kaohsiung Medical University Hospital, with a total of 1,720 beds; Kaohsiung Chang Gung Memorial Hospital, a total of 2,680 beds)...”
Comment 11
Under measures, could you explain a bit more about the three domains of parenting stress? It is not clear in the introduction nor in the methods section what these domains are about.
Response
Thank you for your suggestion. We added more introduction for the three domains of the Parenting Stress Index, Fourth Edition Short Form with example items as below. Please refer to line 167-174.
“The items on the PD domain assess levels of distress resulting from personal factors of caregivers such as depression or conflicts with a partner and life restrictions due to the demands of CADHD rearing (e.g., “I often have the feeling that I cannot handle things very well”). The items on the PCDI domain assess levels of caregivers’ dissatisfaction of interactions with CADHD and the degree to which caregivers find CADHD unacceptable (e.g., “My child rarely does things for me that make me feel good”). The items on the DC domain assess CADHD’s self-regulatory abilities perceived by the caregivers (e.g., “My child seems to cry or fuss more often than most children”).”
Comment 12
Under statistical analysis, may I know how you determined whether there was a moderating effect of demographic characteristics, inattention problems, hyperactive and impulse control problems, and ODD symptoms? Please clarify clearly how the moderation analysis was done, and how the effect of moderation was determined.
Response
We added the introductions for how this study determined the moderators. We also added the introduction for how we presented the moderating effects in a figure. Please refer to line 209-222.
“We also used the standard criteria proposed by Baron and Kenny [35] to examine whether the association between AS and parenting stress were different in terms of demographic characteristics, inattention problems, hyperactive and impulse control problems, and ODD symptoms. According to the criteria, moderation occurred when the interaction term for the predictor (AS) and the hypothesized moderator (demographic characteristics, inattention problems, hyperactive and impulse control problems, and ODD symptoms) were significantly associated with the dependent variable (parenting stress) after controlling for the main effects of both the predictors and hypothesized moderator variables. If AS and hypothesized moderators were significantly associated with parenting stress, the interactions (AS × hypothesized moderators) were further incorporated into the regression analysis to examine the moderating effects. Moderation analysis was performed using the PROCESS v4.0.0 macro, which is based on linear regression modeling and developed for Statistical Analysis Software (SAS) 9.4 (SAS Institute Inc., Cary, NC).”
Comment 13
Results. In Table 2, please indicate which is the reference group, male or female, yes or no, for each of the categorical variables used.
Response
Thank you for your reminding. We indicated the reference group into Table 2 as below. Please refer to line 246.
“a: Females as the reference; b: Girls as the reference.”
Comment 14
Please report the R-squared, adjusted R squared values, and the F statistics of each multiple regression model.
Response
We added the R-squared, adjusted R squared values, and the F statistics of each multiple regression model into Table 2. Please refer to line 244-246.
Comment 15
- 167 - "AS was significantly associated" did you mean "higher AS was significantly associated..."
Response
We added “higher” into this sentence. Please refer to line 238.
Comment 16
l.180, rather than "increased parenting stress", kindly consider changing it to "increased the magnitude of parenting stress". It would be good to indicate how much increase there was.
Response
We changed it into "increased the magnitude of parenting stress" in the revised manuscript. Please refer to line 252-253. We also added Figure 1 to present the magnitude of parenting stress in caregivers with AS moderated by child ODD symptoms. Please refer to Figure 1.
Comment 17
More elaboration on the results for table 3, especially to provide results on by parenting stress domains (the three domains of parental distress, parent-child dysfunctional interaction, and difficult child.
Response
We added elaboration on the results for Table 3. Please refer to line 251-257. We also added Figure 1 to present the moderation effects of ODD symptoms.
“The results revealed that the interaction between ODD symptoms and AS increased the magnitude of parenting stress in the PCDI (coefficient = 0.018, p < 0.05) and DC domains (coefficient = 0.020, p < 0.01), indicating that ODD symptoms moderated the associations between AS and parenting stress in the PCDI and DC domains. The moderating effects are presented in Figure 1. Higher levels of ODD symptoms corresponded to higher parenting stress in the PCDI and DC domains among participants with AS.”
Figure 1. Moderation effects of oppositional defiant disorder (ODD) symptoms on the association between affiliate stigma and parenting stress in caregivers. Mean centering was performed on the predictor (affiliate stigma) and moderator (ODD) symptoms) prior to moderation analysis to enhance comparability.
Comment 18
Discussion. The discussion section is too short for the interpretation to be meaningful. For example, I see that the analysis involved three separate domains of parenting stress; however, this was not discussed in the discussion section. Rather parenting stress was treated as a unidimensional variable. Kindly involve a discussion on how different domains of parenting stress was affected by AS. It is interesting that ODD was not associated with parental distress, but yes for the other two domains of parenting stress. Why do you think that is the case, and how is that different from past literature? Similarly, I would like to suggest discussing the moderating effect of the various factors tested on the three different domains of parenting stress, since different results were obtained for different parenting stress domains.
Response
Thank you for your comment. We revised the contents of Discussion section on how different domains of parenting stress was affected by AS. We also added discussion on the moderating effect of ODD symptoms on the three different domains of parenting stress.
“This study demonstrated that after demographic characteristics and childhood ADHD and ODD symptoms were controlled for, AS was positively associated with all three domains of parenting stress in caregivers of CADHD. This study is the first one to examine the associations between AS and various domains of parenting stress in caregivers of CADHD. Although the cross-sectional design limited the possibility to determine the casual relationship between AS and parenting stress, there can be reciprocal relationship between them. Parenting stress arises when parenting demands exceed the expected and actual resources available to the parents that permit them to succeed in the parent role [13]. Stressful feeling and experiences of failure in parenting will discourage caregivers of CADHD and increase the risk of internalizing public stigma toward ADHD as their attitudes toward their CADHD. Alternatively, AS may increase multiple domains of parenting stress. Caregivers who have higher parenting stress in the PD domain perceived greater distress due to negative emotions, conflicts with family member, and life interferences caused by the demands of parenting [13]. Charbonnier et al. observed that AS was positively correlated with depression and anxiety in mothers of CADHD and was negatively correlated with mothers’ life and self-satisfaction levels [15]. AS may compromise caregivers’ mental health and increase caregivers’ difficulties in cooperating with family members involved in child-rearing; then the parenting stress in the PD domain increases. Caregivers with higher parenting stress in the PCDI domain had greater dissatisfaction with the quality of interactions with their CADHD and greater disapprove toward CADHD’s behaviors [13]. Research found that greater AS was associated with more negative parenting among caregivers of CADHD [23]. AS may reduce caregivers’ patience and ability to learn how to well interact with their CADHD and further exacerbate parent–child dysfunctional interaction. Caregivers with higher parenting stress in the DC domain perceived greater CADHD’s difficulties in self-regulation and fulfillment of their obligations [13]. Studies have found that AS is related to caregivers’ unfavorable attitudes toward clinical diagnoses and treatment for ADHD [19–21]. The timing of proper treatment for ADHD may be delayed, and children’s ADHD symptoms and difficulties in daily lives could be aggravated. Research found that caregivers’ AS predicted CADHD’s emotional problems and physical discomfort [22]; therefore, caregivers’ parenting stress in the DC domain may increase.” Please refer to line 268-298.
“…research found that ODD symptoms of CADHD had various associations with the three domains of parenting stress. Wiener et al. found that mothers of CADHD with ODD reported higher levels of parenting stress than mothers of CADHD without ODD in the DC and PCDI domains and that fathers of CADHD with ODD reported more PCDI domain stress than fathers of CADHD without ODD [3]. The present study also found that higher ODD symptoms of CADHD were significantly associated with parenting stress in the DC and PCDI but not PD domains and further moderated the association between AS and parenting stress in the PCDI and DC domains. Children with ODD symptoms are uncooperative, defiant, and hostile toward caregivers and have low intentions to comply with caregivers’ requests. Therefore, parenting stress in the DC and PCDI domains involving the management of CADHD’s behaviors can increase [25].” Please refer to line 312-323.
Comment 19
Some of the studies cited are really dated. May I know why that is the case?
Response
Thank you for your comment. As mentioned in the response to Comment 7, cited more studies published in recent years in the revised manuscript.
Comment 20
Under limitations, please specify how the generalizability of the study is affected by the recruitment method.
Response
Thank you for your reminding. We added it as one of the limitations as below. Please refer to line 346-349.
“Fourth, this study recruited caregivers of CADHD from the outpatient clinics. Whether the results of this study can be generalized to caregivers who did not take their CADHD to child psychiatrists warrants further study.”
Round 2
Reviewer 3 Report
All the concerns have been addressed well. Thank you.